# Flow Matching for Denoised Social Recommendation

**Yinxuan Huang** [1] [*]  **Ke Liang** [1] [*]  **Zhuofan Dong** [2]  **Xiaodong Qu** [3]  **Tianxiang Wang** [1]  **Yue Han** [1]  **Jingao Xu** [1]
**Bin Zhou** [1]  **Ye Wang** [1]

## Abstract

Graph-based social recommendation (SR) models suffer from various noises in social graphs, hindering their recommendation performances. Both graph-level redundancy and graph-level missing will indeed influence the social graph structures, further influencing the message propagation procedure of graph neural networks (GNNs). Generative models, especially diffusion models, are usually used to reconstruct and recover the data in better quality from noisy input. Motivated by it, a few works take attempts on it for social recommendation. However, they can only handle isotropic Gaussian noise and fail to address anisotropic noise. Moreover, an anisotropic relational structures in social graphs are commonly seen, which existing models cannot sufficiently utilize the graph structures, which constraints the capacity of noise removal and recommendation performances. Compared to the diffusion strategy, the flow matching strategy better handles anisotropic noise, as it preserves data structures more effectively during the learning process. Inspired by this, we propose RecFlow, the first flow-based SR model. Concretely, RecFlow performs flow-based method on the structure representations of social graphs. Then, a conditional learning procedure is designed for optimization. Extensive performances prove the promising performances of our RecFlow from six aspects, including superiority, effectiveness, robustnesses, sensitivity, convergence and visualization. Code are available at 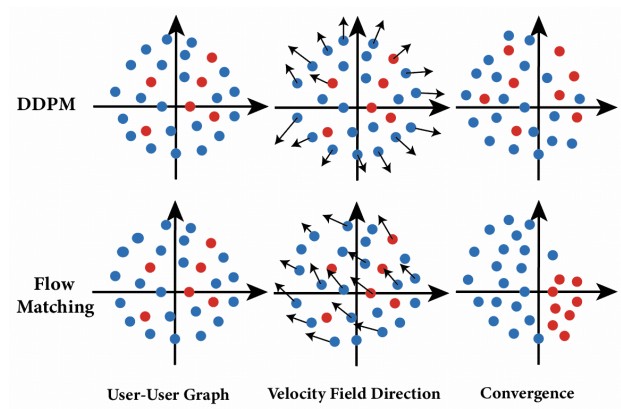.

---

[*]Equal contribution  [1]School of Computer Science, National University of Defense Technology, Changsha, Hunan, China [2]University of Chicago, Chicago, Illinois, United States [3]Harbin Institute of Technology(Shenzhen), Shenzhen, Guangdong, China. Correspondence to: Ye Wang <Yewang@nudt.edu.cn>.

*Proceedings of the $42^{nd}$ International Conference on Machine Learning*, Vancouver, Canada. PMLR 267, 2025. Copyright 2025 by the author(s).

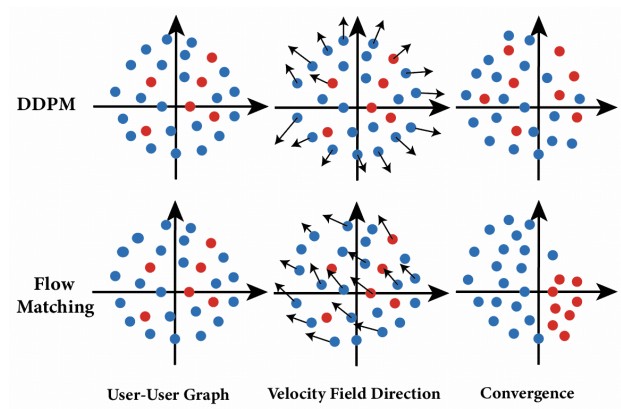

*Figure 1.* Illustration of denoising diffusion probabilistic models (DDPM) and flow matching based models where blue and red nodes represent normal and noisy data. Compared to DDPM, flow-based models can get better discriminative capacity, further leading to better denoise performance.

## 1. Introduction

As online content continues to grow exponentially, the challenges of managing information sensitivity and urgency have become increasingly pressing (Lin et al., 2025; Wang et al., 2024), driving the rise of recommendation systems (Liu et al., 2023b). Despite advancements, recommendation systems still face challenges such as collaborative information sparsity. The rise of social media has shifted their focus from user-item interactions to integrating social networks to enhance recommendations. By leveraging social relationships as auxiliary information, social recommendation (SR) systems can mitigate these issues, making SR a key research area (Liang et al., 2023). Early models of SR relied on matrix factorization (Salakhutdinov & Mnih, 2007; Yang et al., 2013), which drew upon social theories to exploit the influence of nearby or connected users on individual preferences. Additionally, social relationships naturally form graph-structured data, making graph neural networks (GNNs) an effective tool for graph representation learning (Wang et al., 2023a; Dai et al., 2023). GNNs have demonstrated exceptional performance in aggregating neighborhood information of nodes and have been widely

applied in the social recommendation domain, enabling the deep exploration and effective utilization of more valuable information (Huang et al., 2021a; Liang et al., 2023).

Despite advancements in social recommendation systems, current approaches often struggle to effectively mitigate two critical graph structural issues: redundancy and incompleteness (Lin et al., 2023). These limitations primarily stem from noisy social connections in real-world data, where low-quality relationships inject interference that significantly degrades recommendation performance (Lin et al., 2024c;a). Graph neural networks (GNNs) exhibit particular vulnerability to such noise due to their inherent reliance on message propagation mechanisms across social edges, a characteristic that amplifies error transmission through the network (Wang et al., 2023a; Lin et al., 2024b). Recent studies have extended diffusion models to graph-based recommendation systems. DiffRec (Wang et al., 2023b) applies continuous diffusion by adding Gaussian noise to user/item embeddings and optimizes them through a denoising process. RecDiff (Li et al., 2024b) introduces a multi-step diffusion and denoising framework for modeling complex social connections. These efforts build upon the growing success of diffusion models in various domains (Liu et al., 2024; Lee et al., 2024; Jiang et al., 2024; Deng et al., 2024).

Social data often exhibit strong anisotropy, as shown by the directional distribution of vector fields in our preliminary analysis (Figure 3). This conflicts with the **isotropic Gaussian noise** (where noise is modeled as $\epsilon \sim \mathcal{N}(0, \sigma^2 I)$) assumption in conventional denoising diffusion probabilistic models (DDPM), leading to two key issues: representation degradation, as isotropic noise blurs user embedding distinctiveness, and unstable training, as DDPM's iterative denoising undermines convergence consistency (Kingma et al., 2023). To address these limitations, we adopt Flow Matching, a generative approach that learns continuous velocity fields to construct direct sampling paths and model an ODE flow, enabling it to handle **non-isotropic noise**(modeled by $\epsilon \sim \mathcal{N}(\mu, \Sigma)$ with $\mu \neq 0$ or non-diagonal $\Sigma$) (Zhao et al., 2024; Lipman et al., 2022). Figure 1 illustrates the differences between these generative approaches: DDPM injects isotropic noise without directional awareness, while Flow Matching guides data towards clean distributions in a more discriminative manner, improving denoising performance on graph-structured data (Lipman et al., 2022). Building on this, we propose RecFlow, a flow-based social recommendation model that leverages Flow Matching to capture the directional dynamics of user interactions, enabling more stable training, better denoising, and improved recommendation accuracy. Building on this, we propose **RecFlow**, a flow-based social recommendation model that leverages Flow Matching to capture the directional dynamics of user interactions, enabling more stable training, better denoising, and improved recommendation accuracy.

In summary, we make the following contributions:

- We propose RecFlow, a novel social recommendation model that explicitly captures anisotropy in social networks through velocity fields, addressing the limitations of isotropic noise assumptions in conventional diffusion models.

- By integrating flow matching with social recommendation, RecFlow models directional data dynamics more effectively, leading to improved user preference representations.

- Extensive experiments on benchmark datasets demonstrate the effectiveness of RecFlow, with significant performance gains. A visualization experiment further visualizes the evolution of velocity fields over time, highlighting the impact of our approach.

## 2. Related Work

In this section, we provide a comprehensive review of related studies in the areas of social recommendation and generative models, and clarify how our work aligns with and builds upon the existing research.

### 2.1. Graph-based Social Recommendation

Graph-based Social Recommendation has gained significant attention for incorporating social relationships. Early works like DiffNet(Wu et al., 2019b) used Graph Convolutional Networks (GCNs) to model social influence, while later models like GraphRec(Fan et al., 2019) and DANSER(Wu et al., 2019c) added attention mechanisms to account for varying influence levels. More recent approaches, such as MHCN(Yu et al., 2021b) and HOSR(Liu et al., 2020), capture higher-order relationships and distant influences. Models like RecoGCN(Xu et al., 2019), DGRec(Song et al., 2019b), TGRec(Bai et al., 2020), and KCGN(Huang et al., 2021b) integrate diverse data sources, including agent, temporal, and knowledge graph information. To address noisy social relations, recent methods like DSL(Wang et al., 2023a) and GDMSR(Quan et al., 2023) focus on denoising by identifying and removing irrelevant or redundant social connections, thus improving recommendation quality and efficiency. These methods struggle with noisy or irrelevant social connections. Flow matching models, by modeling influence spread and denoising, provide a promising solution, refining user representations to enhance the robustness and accuracy of recommendations.

### 2.2. Diffusion Model-based Recommendation

Generative recommenders have attracted interest in recent studies. Some studies focused on leveraging diffusion mod-

els to enhance data representation and mitigate noise inherent in social connections. For instance, RecDiff(Li et al., 2024a) employed a latent diffusion paradigm to denoise user representations derived from social networks, demonstrating improved robustness in handling the diverse noisy effects of user social contexts. Similarly, DiffuASR(Liu et al., 2023a) proposed a diffusion-based pseudo sequence generation framework, and fills in the gap between the generations of continuous images and discrete sequences. CGSoRec(He et al., 2024) proposed a condition-guided social recommendation model, leveraging a conditional constraint in the diffusion process to incorporate social connections. This allows the model to refine user preferences based on their social connections. Similarly, DIEXRS(Guo et al., 2023) uses a diffusion framework to model user preferences, and then trains a textual decoder to generate explanations based on the denoised user representation, enhancing the interpretability of diffusion recommenders. In contrast to these established approaches, our proposed Recflow introduces a novel methodology by leveraging flow matching models (Liu et al., 2022), making it more effective in capturing intricate patterns.

## 3. Preliminary

In this section, we briefly introduce the preliminaries of flow-matching models. A flow-matching model is a type of generative model that bridges the gap between a source distribution $p_x$ and a target distribution $p_z$ by learning a neural network to parameterize the velocity field of an Ordinary Differential Equation (ODE). The ODE is defined as:

$$dx_t = v_\theta(x_t, t)dt, \qquad (1)$$

where $v_\theta$ denotes the learnable velocity field parameterized by a neural network. The ODE ensures that the intermediate distributions $x_t$ remain consistent with the learned probability path for all $t \in [0, 1]$, and the velocity field $v_\theta$ directs the flow from the initial state $x$ to the target state $z$, effectively transforming the source distribution into the target distribution over time.

**Forward Process:** This process converts samples from $x \sim p_x$ to align with $p_z$. The interpolation between $x$ and $z$ is defined through the linear blend $x_t = tz + (1 - t)x$, satisfying the ODE:

$$dx_t = (z - x)dt \qquad (2)$$

**Reverse Process:** Conversely, this process generates samples starting from $z \sim p_z$ and reverses the flow dynamics. The reverse ODE, mirroring the forward process, is defined as:

$$dx_t = (x - z)dt \qquad (3)$$

The effectiveness of the rectified flow depends on the precise estimation of velocity $v$. To align $v$ with the direction $(z -$

$x)$, the model solves a least squares regression problem, optimizing the velocity field $v_\theta$ to closely match the ideal flow between $x$ and $z$.

The training of the neural network involves minimizing the loss function $L$, defined as:

$$\mathcal{L}_2 = \int_0^1 \mathbb{E}_{x,z} \left[ \|(z - x) - v_\theta(x_t, t)\|^2 \right] dt \qquad (4)$$

This loss quantifies the discrepancy between the ideal and predicted velocities over the time interval $[0, 1]$, enabling the flow to follow the desired trajectory by accurately predicting the velocity at any point $t$. The parameterized neural network $v_\theta$ is thus trained to minimize $L$, facilitating an efficient and accurate modeling of transitions from $x$ to $z$.

## 4. Proposed Model

As illustrated in Figure 2, we integrated collaborative and social signals within a unified flow-based generative framework, the overall architecture of RecFlow consists of three key components: Graph-based Collaborative Pattern Encoding, the RecFlow Module, and a Joint Optimization Module.

### 4.1. Problem Statement

Users and items are defined as the sets $U = \{u_1, u_2, \ldots, u_n\}$ and $V = \{v_1, v_2, \ldots, v_n\}$, respectively. Interactions between users and items are represented by the matrix $R \in \mathbb{R}^{|U| \times |V|}$, where the element $r_{u,v} = 1$ indicates that user $u$ interacts with item $v$, and $r_{u,v} = 0$ otherwise.

Social relationships between users are described by the matrix $S \in \mathbb{R}^{|U| \times |U|}$, where $s_{u,u'} = 1$ signifies a social interaction between user $u$ and user $u'$, and $s_{u,u'} = 0$ otherwise. Based on these interaction matrices, the following graph structures are constructed:

- **Collaborative Graph** $G_r = (U, V, E_r)$, where the edge set $E_r = \{(u, v) \mid r_{u,v} = 1\}$ represents interactions between users and items.

- **Social Graph** $G_s = (U, E_s)$, where the edge set $E_s = \{(u, u') \mid s_{u,u'} = 1\}$ represents social relationships between users.

Our RecFlow leverages both collaborative and social graphs specifically, the collaborative graph $G_r$ generates node embeddings denoted as $E_r$ and the social graph $G_s$ generates node embeddings denoted as $E_s$. The predicted user-item interaction value $\hat{r}_{u,v}$ is computed as:

$$\hat{r}_{u,v} = \text{Pred}(e_u, e_v), \qquad (5)$$

where $e_u$ and $e_v$ are the embeddings of the user $u$ and item $v$. These $e_u$ and $e_v$ are derived from both the collaborative

---

**Algorithm 1** RecFlow Training

---

**Input**: Users' social interaction embedding $E^s$
**Output**: The reconstructed embedding which is denoted as $\hat{e}_0$

 1: Set $E^s$ # Initialize the social interaction embedding
 2: **while** not converged **do**
 3:    $t \sim \mathcal{U}(0,1)$        # Sample time
 4:    $x \sim p_x$        # Sample data
 5:    $z \sim p_z$        # Sample noise
 6:    $x_t = \Psi_t(z|x)$        # Conditional flow
 7:    Gradient step: $\nabla_\theta \| v_t^\theta(x_t) - \hat{x}_t \|^2$
 8: **end while**

---

**Algorithm 2** RecFlow Inference

---

**Input**: Users' interaction vectors $x_u$, $u = 1, 2, \ldots, |U|$; optimized parameter $\theta$
**Output**: Predicted user embeddings or interaction outcomes

 1: **for** $u \in U$ **do**
 2:    Let $x_0 \leftarrow x_u$        # Initialize with user data
 3:    Sample $t \sim p_t$        # Time step sampling
 4:    Calculate $x_t \leftarrow f_\theta(x_0, t)$        # Run learned model
 5:    **if** needsPostProcessing **then**
 6:       $\hat{y} \leftarrow \text{Process}(x_t)$        # Final prediction
 7:    **end if**
 8: **end for**

---

graph $G_r$ and the social graph $G_s$, and are learned jointly during the model training, respectively.

## 4.2. Graph-based Collaborative Pattern Encoding

Drawing inspiration from the effectiveness of simplified Graph Neural Networks (GNNs), we incorporated a lightweight Graph Convolutional Network (lightGCN) as the graph encoder in our RecFlow architecture (Jiang et al., 2023). LightGCN is widely recognized as a robust graph recommender for modeling implicit interactions in top-k recommendations. we construct a collaborative graph to encode user-item interactions using the Rec Encoder, and a user-user graph to capture social relationships among users using the Social Encoder. On the user-item graph $G_r$, the embeddings are propagated across layers using the following equation:

$$
\begin{aligned}
E_r^{(l)} &= (L_r + I) \cdot E_r^{(l-1)}, \\
L_r &= D_r^{-\frac{1}{2}} A_r D_r^{-\frac{1}{2}}.
\end{aligned}
\tag{6}
$$

where $A_r \in \mathbb{R}^{(|\mathcal{U}|+|\mathcal{V}|) \times (|\mathcal{U}|+|\mathcal{V}|)}$ is the adjacency matrix of the bipartite graph $G_r$, the embedding matrix $E_r^{(l)} \in \mathbb{R}^{(|\mathcal{U}|+|\mathcal{V}|) \times d}$ captured the embeddings in the l-th iteration of the GCNs. And $L_r$ is the normalized Laplacian matrix.

The initial embeddings $E_0^r$ are randomly generated learnable parameters. And $D_r$ is the corresponding diagonal degree matrix, defined as:

$$
A_r = \begin{bmatrix} 0 & R \\ R^\top & 0 \end{bmatrix}.
\tag{7}
$$

where $L_r$ is the normalized Laplacian matrix of $G_r$, and $I$ is the identity matrix.

For the user's social graph $G_s$, embedding propagation follows a similar process:

$$
\begin{aligned}
E_s^{(l)} &= (L_s + I) \cdot E_s^{(l-1)}, \\
L_s &= D_s^{-\frac{1}{2}} S D_s^{-\frac{1}{2}}.
\end{aligned}
\tag{8}
$$

$S$ being the adjacency matrix of the social graph $G_s$. $L_s$ is the normalized Laplacian matrix. After propagating through $L$ layers, the final embedding for each user-item pair is obtained by aggregating embeddings across all layers as $\hat{e}_{u,u'} = \sum_{l=0}^{L} e_{u,u'}^{(l)}$, where $e_{u,u'}^{(l)}$ represents the $l$-th layer embedding between user $u$ and $u'$. Similarly, for a user-item pair $(u, v)$, the predicted embedding is $\hat{e}_{u,v} = \sum_{l=0}^{L} e_{u,v}^{(l)}$.

## 4.3. RecFlow Module

In the forward process, the RecFlow module begins by sampling a time step $t$ from a uniform distribution $\mathcal{U}(0, 1)$. This time step is then encoded into a time embedding vector $E_t$, which is concatenated with the user representation $E_s$ obtained from the social encoder. The perturbed input $x_t$ is defined as a linear interpolation between Gaussian noise $z$ and the socially-informed embedding $E_s$:

$$
x_t = tz + (1 - t)E_s
\tag{9}
$$

Unlike conventional flow-matching methods that typically treat $z$ as the origin and interpolate toward data samples, our formulation conditions the diffusion trajectory on the social embedding $E_s$, enabling the model to incorporate social context into the forward process.

In the reverse process, we train a vector field estimator $v_\theta(x_t, t)$ to approximate the velocity field. Here, $\theta$ denotes the set of learnable parameters. The estimator is defined as:

$$
v_\theta(x_t, t) = \text{FC}^2(E_s \,\|\, E_t), \quad \text{FC}(x) = \sigma(Wx + b)
\tag{10}
$$

where $E_t$ is the time embedding at step $t$, $\|$ denotes vector concatenation, and $\text{FC}^2$ represents two consecutive fully connected layers. The function $\sigma(\cdot)$ denotes a non-linear activation (e.g., ReLU or GELU), and $W$, $b$ are the weight matrix and bias vector of each linear transformation.

Specifically, starting from $x_t$, the model integrates the reverse-time flow using an ODE solver to obtain the clean

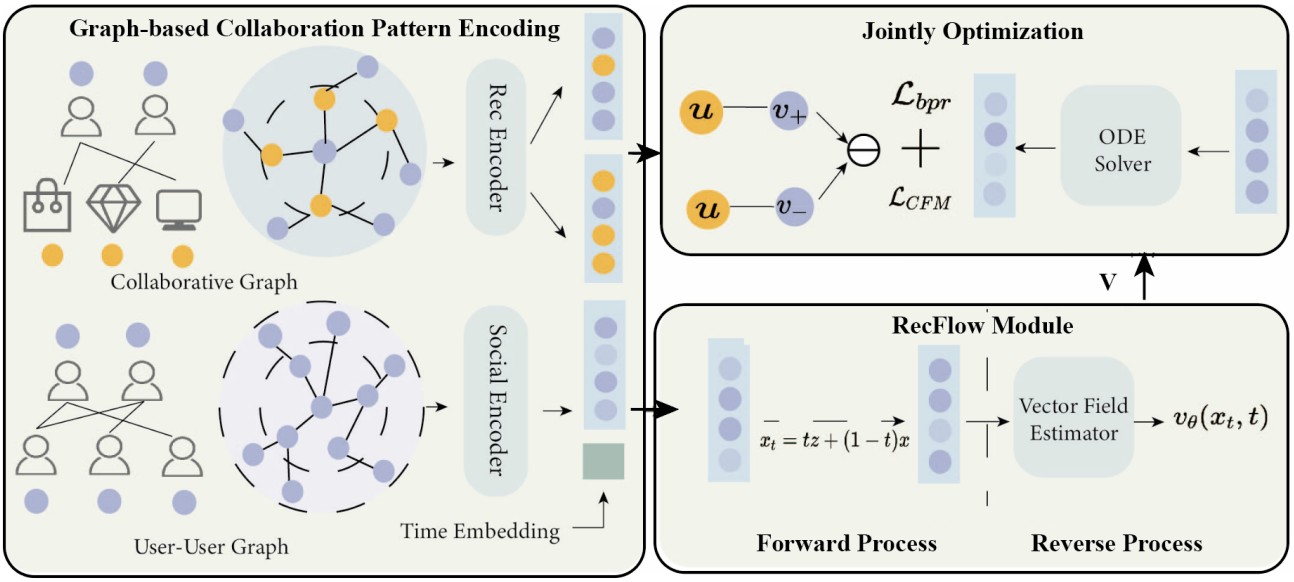

*Figure 2.* Our RecFlow consists of three main components: **Graph-based Collaborative Pattern Encoding** responsible for obtaining embeddings of the collaborative graph and the social graph; **RecFlow Module** learns continuous velocity fields on the social graph to effectively denoise anisotropic structures; and **Joint Optimization** integrates the objectives of collaborative encoding and flow-based social representation learning.

embedding $x_0$, which approximates the original user preference vector in the latent space. The reverse integration is performed as:

$$x_0 = x_t + \int_t^0 v_\theta(x_\tau, \tau)\,d\tau \qquad (11)$$

During inference, the model fixes the optimized parameters $\theta$, and no further training is performed, then we sample a latent representation $z \sim \mathcal{N}(0, I)$ and timestep $t$ and then applies the learned $v$ to reconstruct the user embedding $e_u$.

### 4.4. Joint Optimization

To integrate social relationships with encoded user-item interaction patterns, RecFlow employs a hidden-space reflow mechanism to generate the final user embeddings for prediction. This process is defined as:

$$\hat{r}_{u,v} = \tilde{e}_u^\top e_v', \quad \tilde{e}_u = e_u' + \hat{e}_\theta(e_u', t), \qquad (12)$$

where $t$ denotes a sampled diffusion time step for user $u$, $e_u'$ and $e_v'$ represent the initial embeddings of the user and item obtained from the respective graph encoders, and $\hat{e}_\theta(\cdot, t)$ denotes the learned reflow adjustment from the vector field estimator.

The model is optimized by minimizing a joint loss function that combines recommendation and diffusion objectives:

$$\mathcal{L} = \sum_{(u,v^+,v^-)} -\log \sigma(\hat{r}_{u,v^+} - \hat{r}_{u,v^-}) + \lambda_1 \sum_t \mathcal{L}_{\text{cfm}} \quad (13)$$

Here, $(u, v^+, v^-)$ denotes a user with a positive and a negative item in a pairwise training setup following the Bayesian Personalized Ranking (BPR) paradigm (Rendle et al., 2012). The conditional flow-matching loss $\mathcal{L}_{\text{cfm}}$, computed over sampled diffusion steps $t$, guides the learning of the reverse-time vector field. Additionally, L2 regularization (weight decay) with coefficient $\lambda_1$ is applied to all trainable parameters $\Theta$ to prevent overfitting. The form of $\mathcal{L}_{\text{CFM}}$ follows the definition in Eq. (3) and the flow estimation process illustrated in Figure 2.

### 4.5. Disucssion and Analysis

In this section, we present a detailed analysis of the time and space complexity of our RecFlow model. Besides, we further provide the theoretical analysis between our RecFlow and original DDPM methods to better illustrate the efficiency of the flow matching strategy.

**Time Complexity.** Initially, RecFlow performs graph-level information propagation on both the holistic collaborative graph $\mathcal{G}_r$ and the social graph $\mathcal{G}_s$, this process requires $\mathcal{O}((|\mathcal{E}_r| + |\mathcal{E}_s|) \times d)$ calculations for message passing. The overall time complexity of RecFlow during training is dominated by the graph-level propagation and the gradient updates, resulting in:

$$\mathcal{O}((|\mathcal{E}_r| + |\mathcal{E}_s|) \times d) \qquad (14)$$

for each iteration of training. And complexity of flow match-

ing is O(N), eliminating multi-step iterations and requiring only one global optimization.

**Space Complexity.** The space complexity is primarily determined by the storage required for the graphs and embeddings. The collaborative graph $\mathcal{G}_r$ requires storing the adjacency matrix of the user-item interactions, which has a space complexity of $\mathcal{O}(|U| \times |V|)$, the social graph $\mathcal{G}_s$ requires storing the adjacency matrix of the user-user interactions, which has a space complexity of $\mathcal{O}(|U|^2)$, assuming a dense representation. Thus, the overall space complexity of RecFlow is:

$$\mathcal{O}(|U| \times |V| + |U|^2 + (|U| + |V|) \times d) \quad (15)$$

This accounts for the storage of the graph structures and the user and item embeddings. And space complexity is O(D), dependent solely on data dimensionality and independent of timesteps.

**Theoretical Analysis.** In the context of generative modeling, **Flow Matching** and **DDPM** both aim to generate data through controlled transformations of noise. FM constructs a linear interpolation between $X_0$ and $X_1$, leading to a continuous and deterministic path. Mathematically, this is described as:

$$X_t = (1 - t)X_0 + tX_1 \quad (16)$$

where $X_t$ follows a simple ODE-driven trajectory. In contrast, DDPM follows a stochastic diffusion process:

$$X_t = \sqrt{\alpha_t}X_0 + \sqrt{\beta_t}\epsilon \quad (17)$$

where $\epsilon$ is sampled from a Gaussian prior. The nonlinear and stochastic nature of DDPM results in higher variance in sampling paths, leading to inefficiencies. And FM is governed by an Ordinary Differential Equation (ODE), which enables continuous time evaluation and faster sampling via adaptive solvers. In contrast, DDPM relies on discrete Stochastic Differential Equations (SDEs), requiring a large number of steps for accurate generation. Empirically, FM achieves comparable quality with fewer function evaluations, reducing computational overhead.

## 5. Experiment

In this section, we present a series of experiments conducted to evaluate the performance of our RecFlow method, focusing on the following five questions:

- **Q1:** How does RecFlow perform in comparison to other state-of-the-art social recommendation methods?

- **Q2:** What are the key contributions of RecFlow's main modules?

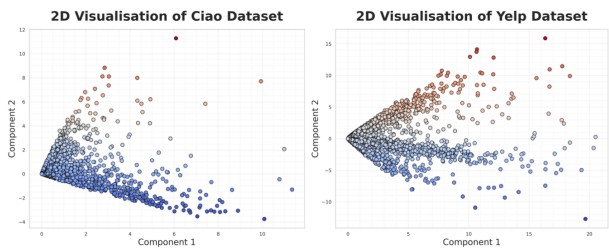

*Figure 3.* Illustration of the anisotropic attributes of two typical datasets, i.e., Ciao and Epinions. Each point in the 2D visualization represents a user–user interaction embedding projected onto two principal components. The uneven and elongated distributions of points along certain directions highlight the anisotropy of the data, indicating that user–user relationships are not uniformly distributed but instead exhibit directional concentration.

- **Q3:** Is RecFlow robust enough to effectively handle noisy and sparse data in social recommendation (SR)?

- **Q4:** How do different settings impact the performance of RecFlow?

- **Q5:** How does the complexity of our method compare to that of alternative approaches?

Before showing and analyzing the experimental results, we first present the experimental settings below.

### 5.1. Experiment Settings

**Datasets and Evaluation Metrics.** We conducted experiments on three publicly available social recommendation datasets: Ciao, Yelp and Epinions. Detailed statistics for these datasets are provided in Table 2. Our datasets are obtained from RecDiff (Li et al., 2024a)[1]

We conducted a preliminary analysis of the Ciao and Epinions dataset. Figure.3 shows the distribution of social data in the Ciao and Epinions dataset. After reducing the data from high-dimensional space to two dimensions using Principal Component Analysis (PCA), it can be observed that the data points are more spread out along Component 1, while the distribution is more concentrated and less variable along Component 2. This aligns with the characteristics of social network data, where there are often dominant relational patterns, while others are secondary or sparse. In the social graph, certain user groups with strong internal connections form tightly-knit social clusters, reflecting prominent relational patterns in the data. Additionally, the presence of a dominant direction in the data causes the vector field to display a clear anisotropic distribution, indicating that user interactions are more concentrated along specific directions.

---

[1]Datasets are available at https://github.com/HKUDS/RecDiff.

*Table 1.* Statistics of experimental datasets, our datasets

| Data | Ciao | Yelp | Epinions |
|---|---|---|---|
| # Users | 1,925 | 99,262 | 14,680 |
| # Items | 15,053 | 105,142 | 233,261 |
| # Interactions | 23,223 | 672,513 | 447,312 |
| # Social Ties | 65,084 | 1,298,522 | 632,144 |

**Evaluation Protocols.** In our experiments, we utilized two commonly used metrics.: Hit Ratio HR@N and Normalized Discounted Cumulative Gain (NDCG)@N as metrics, where $N$ represents the number of items recommended to the user, widely ultilized in Top-N recommendations. We applied a 7:1:2 ratio to split each dataset into training, validation, and test sets, adhering to common data partitioning practices in graph-based recommendation systems.

**Compared Baselines.** We compared RecFlow with 12 baseline models, representing the latest advancements in social recommendation research. These encompass conventional and attention-based methods, graph-based recommendation models using collaborative filtering, as well as other GNN-based social recommendation systems, as outlined below: PMF (Salakhutdinov & Mnih, 2007), TrustMF (Yang et al., 2013), GraphRec (Fan et al., 2019), DiffNet (Wu et al., 2019a), DGRec (Song et al., 2019a), NGCF (Wang et al., 2019), MHCN (Yu et al., 2021a), KCGN (Huang et al., 2021a), SMIN (Long et al., 2021), GDMSR(Quan et al., 2023), DSL (Wang et al., 2023a), RecDiff (Li et al., 2024a).

**Implementation Details.** All experiments are conducted on a machine with an RTX A800 for a fair comparison. The experimental settings and hyper-parameters details of our RecFlow framework are elaborated in. The learning rate was tuned within $[5e^{-4}, 1e^{-3}, 5e^{-3}]$ with a 0.96 decay factor per epoch. Batch sizes were selected from [1024, 2048, 4096, 8192], and hidden dimensions from [64, 128, 256, 512]. The parameter $\gamma$ was set according to the $\gamma_{pct}$-percentile of node embedding distances for each dataset. The optimal number of GNN layers was chosen from [1, 2, 3, 4]. The Timestep embedding size is selected from 4,8,16,32. And the batch size for Ciao is 2048, while for Yelp and Epinions is 4096. Regularization weights $\lambda_1$ were selected from $[1e^{-3}, 1e^{-2}, 1e^{-1}, 1e^0, 1e^1]$.

### 5.2. Performance Comparison (RQ1)

Table 2 summarizes the experimental results across three datasets, with RecFlow's metrics bolded and top baselines underlined. RecFlow demonstrates marked improvements over existing approaches, more specifically, on Ciao, RecFlow achieves 0.725 Recall (+2.0%) and 0.438 NDCG

*Table 2.* Overall performance analysis.

| Method | Ciao Recall | Ciao NDCG | Yelp Recall | Yelp NDCG | Epinions Recall | Epinions NDCG |
|---|---|---|---|---|---|---|
| TrustMF | 0.539 | 0.343 | 0.371 | 0.193 | 0.265 | 0.195 |
| SAMN | 0.604 | 0.384 | 0.403 | 0.208 | 0.329 | 0.226 |
| DiffNet | 0.528 | 0.328 | 0.557 | 0.292 | 0.384 | 0.273 |
| GraphRec | 0.540 | 0.335 | 0.419 | 0.201 | 0.334 | 0.246 |
| DGRec | 0.517 | 0.319 | 0.410 | 0.209 | 0.326 | 0.236 |
| NGCF | 0.559 | 0.363 | 0.450 | 0.230 | 0.353 | 0.243 |
| MHCN | 0.621 | 0.378 | 0.567 | 0.292 | 0.438 | 0.321 |
| KCGN | 0.602 | 0.350 | 0.460 | 0.234 | 0.2201 | 0.1456 |
| SMIN | 0.588 | 0.354 | 0.485 | 0.251 | 0.333 | 0.228 |
| GDMSR | 0.560 | 0.355 | 0.513 | 0.246 | 0.368 | 0.241 |
| DSL | 0.606 | 0.389 | 0.504 | 0.259 | 0.365 | 0.267 |
| RecDiff | 0.712 | 0.419 | 0.597 | 0.308 | 0.460 | 0.336 |
| **RecFlow** | **0.725** | **0.438** | **0.618** | **0.341** | **0.486** | **0.341** |

(+4.5%) over RecDiff. For Yelp and Epinions, it maintains robust gains: 0.618 vs. 0.597 Recall(+3.5%) and 0.341 vs. 0.308 NDCG (+10.7%) on Yelp; 0.486 vs. 0.460 Recall (+ 5.6%) on Epinions, demonstrating adaptability to varying data densities. Notably, methods with self-supervised learning (SSL)—MHCN (local-global contrast), KCGN, SMIN (hierarchical relations), and DSL (predictive consistency)—consistently outperform traditional approaches. SSL mitigates noise propagation and interaction sparsity by extracting latent relational patterns, enabling stable representation learning. RecFlow's denoising process, guided by a velocity field, directly optimizes trajectories toward clean data distributions, suppressing noise during refinement. This explains its 5.6–10.7% improvements over diffusion-based RecDiff on sparse datasets.

### 5.3. Ablation Study (RQ2)

To evaluate the impact of different components in the RecFlow framework, we performed an ablation study using three benchmark datasets: Ciao, Yelp, and Epinions. The results are presented in Table 4.

- **w/o Flow**: This configuration excludes the holistic flow-matching module, leaving only the GNN for learning user-item and social relations. As shown in Table 4, the absence of the flow module results in a significant decrease in both Recall and NDCG across all datasets. Specifically, Recall drops by approximately 13% (Ciao), 7% (Yelp), and 12% (Epinions), while NDCG decreases by about 13% (Ciao), 16% (Yelp), and 31% (Epinions). This emphasizes the importance of the denoising mechanism in our model.

- **w/o CL**: In this configuration, we remove the conditional learning (CL) guidance for flow matching. The

*Table 3.* Comparison of different sampling methods

| Method | Ciao | | Yelp | | Epinions | |
|---|---|---|---|---|---|---|
| | Recall | NDCG | Recall | NDCG | Recall | NDCG |
| **ODE-Solver** | 0.725 | 0.438 | 0.618 | 0.341 | 0.486 | 0.341 |
| **Multistep Heun** | 0.710 | 0.411 | 0.594 | 0.322 | 0.460 | 0.325 |
| **RK4** | 0.699 | 0.403 | 0.590 | 0.317 | 0.453 | 0.320 |
| **Heun** | 0.683 | 0.399 | 0.581 | 0.319 | 0.449 | 0.318 |
| **Euler** | 0.670 | 0.383 | 0.570 | 0.311 | 0.438 | 0.305 |

*Table 4.* Ablation Analysis: "w/o Flow" denotes the removal of the RecFlow module, "CL" denotes the removal of conditional learning, and "both" indicates the removal of both components simultaneously.

| Method | Ciao | | Yelp | | Epinions | |
|---|---|---|---|---|---|---|
| | Recall | NDCG | Recall | NDCG | Recall | NDCG |
| **RecFlow** | 0.725 | 0.438 | 0.618 | 0.341 | 0.486 | 0.341 |
| **RecFlow w/o Flow** | 0.633 | 0.380 | 0.573 | 0.301 | 0.429 | 0.297 |
| **RecFlow w/o CL** | 0.692 | 0.401 | 0.597 | 0.320 | 0.443 | 0.312 |
| **RecFlow w/o CL of both** | 0.621 | 0.407 | 0.589 | 0.312 | 0.417 | 0.302 |

results show a notable performance decline, particularly in NDCG across all datasets, with a decrease of around 8% (Ciao), 11% (Yelp), and 12% (Epinions). This demonstrates the critical role of **C**onditional **L**earning in improving model accuracy.

- **w/o both**: In this case, both the flow module and the CL label guidance are removed. The performance experiences a significant drop, with Recall decreasing by about 14% (Ciao), 5% (Yelp), and 7% (Epinions), and NDCG dropping by approximately 7% (Ciao), 9% (Yelp), and 15% (Epinions). This highlights the essential contributions of both the flow-matching process and label guidance in enhancing the model's ability to learn effective user-item and social relationships.

The choice of sampling method creates a clear trade-off between computational cost and model accuracy (see Table 3): the basic Euler method, with its single first-order step, yields the lowest Recall and NDCG, while Heun's two-stage predictor–corrector boosts both metrics modestly at only twice the cost. Leveraging history, the multistep Heun scheme further raises performance, and the four-stage RK4 delivers similar gains by reducing local error through fourth-order updates. Finally, an adaptive ODE solver such as Dormand–Prince, which dynamically adjusts its step size to satisfy error tolerances, consistently achieves the highest Recall and NDCG on Ciao (and likewise leads on Yelp and Epinions), demonstrating that, when resources allow, more precise integration yields the strongest quality.

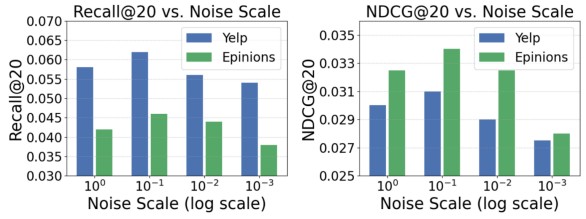

*Figure 4.* Robustness Analysis, the blue bars represent the Yelp dataset, while the green bars represent the Epinions dataset.

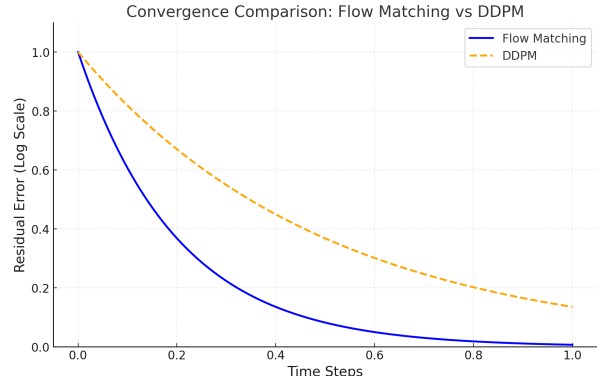

*Figure 5.* The convergence comparison of the loss curves is shown in the legend. The blue curve represents the convergence of RecFlow, while the yellow curve corresponds to the convergence after applying DDPM.

### 5.4. Robustness Analysis (RQ3)

This section examines the effect of the noise scale factor ($\tau$) on the noising process. By scaling the minimum and maximum noise in the scheduler to $\tau \cdot \bar{s}_{\min}$ and $\tau \cdot \bar{s}_{\max}$, respectively, we test the model's performance at different noise scales (1, 0.1, 0.01, 0.001). The results, shown in Figure 4, reveal the following:

- **Increasing the noise scale** improves model performance, with higher Recall@20 and NDCG@20 values for both Yelp and Epinions as the noise scale decreases from 1 to 0.1. This demonstrates the effectiveness of the RecDiff framework's denoising mechanism.

- **Excessive noise beyond a threshold** leads to performance degradation, especially for Yelp and Epinions. As noise scales reach $10^{-2}$ and $10^{-3}$, a noticeable decline in NDCG@20 suggests that too much noise interferes with the model's ability to retain important user-item data.

### 5.5. Convergence Analysis (RQ4)

In Figure.5, RecFlow exhibits a faster convergence speed compared to DDPM, as the horizontal axis represents the

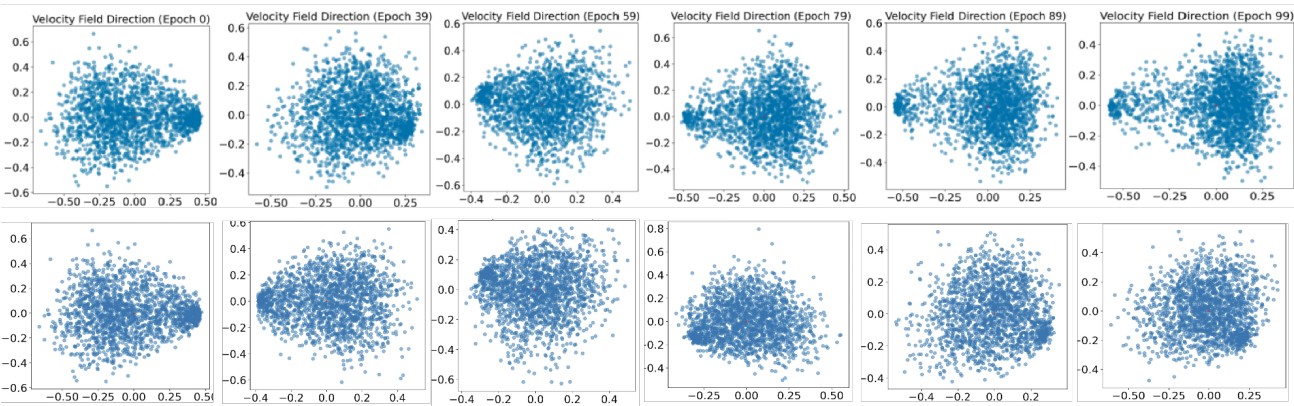

*Figure 6.* The top row shows the velocity field direction of our RecFlow, while the bottom row displays that of DDPM. It is evident that applying the flow-matching velocity field results in a clear directional pattern, aligning with the anisotropy observed in the previous dataset visualizations. In contrast, the application of DDPM does not exhibit any noticeable directionality, with the directions appearing random throughout.

diffusion model's time steps, ranging from 0 to 1. The vertical axis indicates the residual error at each time step. It is evident that the residual error of Flow Matching decreases rapidly in the early time steps, demonstrating a significantly faster convergence trend. In contrast, DDPM's error decreases at a slower pace, highlights the advantage of Flow Matching in modeling vector fields and achieving efficient convergence, making it more suitable for handling complex distributions and data scenarios.

### 5.6. Visualization of Flow matching velocity field direction over different epochs (RQ5)

To better understand how the velocity field evolves during training, we design an experiment that visualizes the direction of the vector field across different epochs. In Figure 6, the data points are unevenly distributed, showing clear directionality and concentration, which highlights the anisotropy of social data. The model captures this anisotropy within the velocity field, represented by arrows that dynamically adjust direction as the epoch evolves, demonstrating the convergence dynamics of velocity field directions.

In the early stages (Epoch 0–38), directions exhibit high dispersion (-0.50 to +0.50), reflecting unstable parameter adjustments. As training progresses (Epoch 59–99), directions progressively cluster near the origin (range: -0.25 to +0.25), indicating stabilized optimization trajectories. Notably, after Epoch 78, data density intensifies with directional similarity, signifying unified parameter updates.

In comparison, as shown in the bottom row of Figure 6, the velocity field direction of DDPM lacks any noticeable directionality across epochs, with the directions appearing random throughout. This contrast highlights that while

RecFlow effectively captures and aligns with the anisotropy of the data, DDPM fails to exhibit clear directional convergence, further emphasizing the advantage of RecFlow in learning meaningful patterns from the social graph structure.

## 6. Conclusion

In this paper, we proposed RecFlow, a novel generative social recommendation framework based on flow-matching. Diffusion-based method primarily target the removal of isotropic noise, therefore damage the structural representation, and the aim of Recflow is to fill the gap. By learning continuous velocity fields on social graphs, Recflow constructs a direct sampling path from noisy input to the target distribution, thereby enabling faithful representation learning. To validate the effectiveness, we conduct extensive experiments on several strong baselines, and the results consistently demonstrate the superior performance and robustness of our proposed RecFlow. In the future, we will explore more scalable extensions of RecFlow to further assess its practicality and effectiveness in real-world recommendation datasets.

## Impact Statement

This paper introduces RecFlow, a flow-matching social recommendation model that captures anisotropic in user interactions. By leveraging flow matching, RecFlow enhances representation learning and denoising efficiency, emphasizes the practical benefits for personalized recommendation. We also acknowledge potential societal risks, such as bias amplification, and highlight the need for fairness and robustness in future recommendation systems.

## Acknowledgments

This work is supported by National Natural Science Foundation of China ( No .62302507) and Hunan Provincial Natural Science Foundation (No. 2023JJ40684).

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
