# OpenReview forum: "Flow Matching for Denoised Social Recommendation"
_ICML.cc/2025/Conference — ICML 2025 poster_

### Official Review · Reviewer_ZxUy · 2025-03-07

**Overall Recommendation:** 4

**Summary:**

1. While there have been many prior works on generative recommendation systems, few have explored the direction of noise. This paper addresses the challenges posed by noisy social networks.
2. It provides a detailed theoretical explanation of the advantages of the flow-matching model, particularly in comparison to DDPM.
3. The paper also includes comprehensive experiments that demonstrate the various strengths of the model.
 In a word, RecFlow effectively addresses the anisotropic noise problem in social recommendation through flow-matching, outperforming existing methods in terms of various perspectives.

**Claims And Evidence:**

1. Could you explain how redundancy differs from errors in this context?
2. I would appreciate a more detailed explanation of how generative models outperform traditional denoising models, if possible, could you provide a small experiment that demonstrates this in action?

**Essential References Not Discussed:**

NA

**Ethical Review Flag:**

Flag this paper for an ethics review.

**Ethics Expertise Needed:**

["Discrimination / Bias / Fairness Concerns"]

**Experimental Designs Or Analyses:**

The theoretical section is already presented quite thoroughly, but the experimental section needs to align with the time complexity discussed in the theory. It would be helpful to directly compare the theoretical time complexity with the experimental results to validate the claims.

**Methods And Evaluation Criteria:**

NA

**Other Comments Or Suggestions:**

NA

**Other Strengths And Weaknesses:**

1. The paper is well-structured, with each section clearly explained, and the theoretical content is presented rigorously.
2. However, the discussion of the problem itself could be further developed. The Introduction section should provide an explanation of the theory behind social homogeneity. While social homogeneity is widely accepted in the social graph domain, it is still important to explicitly address and clarify this concept within the paper.

**Questions For Authors:**

NA

**Relation To Broader Scientific Literature:**

NA

**Theoretical Claims:**

I would appreciate a more detailed explanation of the ODE sampling method and the conditions mentioned in the model section.

---

> ### Author Rebuttal · Authors · 2025-04-01
>
> **1. Differences between redundancy and errors**
> Thanks! Redundancy refers to similar suggestions that do not provide additional value. Errors, on the other hand, represent inaccurate or flawed information, which could arise from noisy data sources, incorrect predictions, or misclassifications. While redundancy can reduce model efficiency by introducing unnecessary information, errors can degrade model accuracy and lead to incorrect recommendations.
>
> **2. More detailed explanation about using generative models rather than traditional denosing models.**
> Thanks! We have already make a comparison to some tranditional denoising models like DSL [1], and the results show that generative models should ideally provide more accurate and diverse predictions because it can better capture the patterns in the noisy data.
> | Model    | Ciao Recall | Ciao NDCG | Epinions Recall | Epinions NDCG |
> |----------|-------------|-----------|-----------------|---------------|
> | GDMSR    | 0.560       | 0.355     | 0.368           | 0.241         |
> | DSL      | 0.606       | 0.389     | 0.365           | 0.267         |
> | RecFlow  | 0.725       | 0.438     | 0.486           | 0.341         |
>
> [1] Denoised Self-Augmented Learning for Social Recommendation
>
>
> **3.ODE sampling method.**
> Thanks! We agree, and we present it as follows, which will also be added in our revised paper.
> Euler method:
> $$ y_{n+1} = y_n + h f(t_n, y_n) $$
>
> Runge-Kutta method:
> $$ k_1 = h f(t_n, y_n), \quad k_2 = h f(t_n + \frac{h}{2}, y_n + \frac{k_1}{2}) $$
> $$ k_3 = h f(t_n + \frac{h}{2}, y_n + \frac{k_2}{2}), \quad k_4 = h f(t_n + h, y_n + k_3) $$
> $$ y_{n+1} = y_n + \frac{1}{6}(k_1 + 2k_2 + 2k_3 + k_4) $$
>
> **4.Condition and supervisory signal**
> The conditions mentioned refer to the labels. These labels guide the interpretation and application of the noise model, defining the structure and nature of the noise at each step. They play a crucial role in conditioning the system, ensuring that the model effectively accounts for variations in the data. In the revised version, we will further elaborate on their role, explaining how these labels influence the model's behavior under different scenarios and how they help in refining the noise removal process for more accurate predictions.
>
>
> **5. Time complexity**
> Thanks! The theoretical time complexity has been discussed in the model module of the paper. In the revised version, we will also include a comparison of the training time to provide a clearer understanding of the model's efficiency.
> | Model    | Ciao | Epinions |
> |----------|------|----------|
> | RecDiff  | 4.5s | 9.1s     |
> | RecFlow  | 4.3s | 8.7s     |
>
>
> **6. Social homogeneity.**
> Thanks! We agree. Following most works [1][2], we also define the isotropy as noise obeys the standard normal distribution N(0,1), which leads to the uniform properties in all directions. While, anisotropy refers to noise distributions that deviate from such uniformity. For example, Gaussian noise with a non-diagonal covariance matrix introduces variability across different directions in the data space. While social homogeneity is widely accepted in the social graph domain, we will explicitly address and clarify this concept within the revised version, relating it to the behavior of isotropic and anisotropic noise in social graphs.
> [1]A Directional Diffusion Graph Transformer for Recommendation
> [1]Denoising Diffusion Probabilistic Models

---

> > ### Comment · Reviewer_ZxUy · 2025-04-02
> >
> > I thank the authors for their response, I will raise my score.

---

### Official Review · Reviewer_avbL · 2025-03-09

**Overall Recommendation:** 3

**Summary:**

The paper introduces a generative model for social recommendation systems, using flow-matching to efficiently handle noise in social networks while preserving relational structures. It provides a thorough theoretical analysis and experiments demonstrating its superiority, fast convergence, and better fitting performance across various data types.

**Claims And Evidence:**

1. Is the denoising process described in the paper consistent with the denoising objective of generative models?
2. Given that noise can vary in type and intensity, how does the paper address these different objectives?
3. Do diffusion models have the ability to differentiate between different types of noise?
4. After denoising, can the representation align with the collaborative signals?

**Essential References Not Discussed:**

All discussed yet.

**Experimental Designs Or Analyses:**

1. Since the theoretical section analyzes how flow-matching methods have lower time complexity compared to DDPM-based methods, it would be useful to compare their time complexities in the experimental section.
2. Regarding convergence analysis, for recommendation tasks, in addition to the convergence curves, it would be helpful to compare recall rate metrics as well.

**Methods And Evaluation Criteria:**

NA

**Other Comments Or Suggestions:**

1. If different sampling methods are utilized, they should be compared in the experimental section.
2. Also, is the coordinate system in the introduction distorted?

**Other Strengths And Weaknesses:**

1. The figure in Introduction only reveals the direction of the arrows when zoomed in, which adds an extra reading burden for the reader.
2. The color scheme of the figures in the paper is inconsistent, and it doesn't align well with the main content. Some explanatory notes should be added to enhance the readability of the article.

**Questions For Authors:**

Applying generative models on the user graph and on the product graph, as in Jiang, Y., Yang, Y., Xia, L., & Huang, C. (2023). *DiffKG: Knowledge Graph Diffusion Model for Recommendation*. arXiv preprint arXiv:2312.16890. https://arxiv.org/abs/2312.16890, as well as applying generative models on the bipartite graph—could these approaches form a unified generative framework for graph-based recommendations? Have you made any attempts in this direction?

**Relation To Broader Scientific Literature:**

No.

**Theoretical Claims:**

1. A more detailed explanation of the flow-matching process is needed.
2. The advantages mentioned in the paper seem to be primarily focused on the flow-matching method itself, rather than specifically in the context of social recommendation.
3. How does flow matching specifically enhance performance within social recommendation scenarios? More clarification is required to better understand its application and benefits in this domain.

---

> ### Author Rebuttal · Authors · 2025-04-01
>
> **1.Consistent with the denoising objective ?**
> Thanks! Our denoising process aligns with the general denoising objective in diffusion models. While typical diffusion models use stochastic noise processes, our method is based on ODEs, achieving a more direct reconstruction objective.
>
> **2.Types of noise?**
> Thanks！While we acknowledge that other types of noise (e.g., label noise, varying intensities) exist, this paper primarily emphasizes modeling anisotropic edge noise. However, addressing various noise types systematically remains a valuable direction for future work. Generally, traditional diffusion models using isotropic Gaussian noise treat all noise uniformly, which limits their ability to explicitly differentiate between noise types. In contrast, our flow-matching approach supports customized anisotropic noise modeling, implicitly differentiating between noise types by adapting noise patterns to relational data structures.
>
> **3.After denoising, can the representation align with the collaborative signals?**
> Yes, the representations obtained after our flow-matching denoising procedure explicitly capture relational (collaborative) structures. By modeling anisotropic noise, our approach preserves critical collaborative signals, resulting in meaningful and effective representations that align closely with recommendation tasks.
>
> **4.A more detailed explanation of the flow-matching process is needed.**
> Thanks！We will include a more detailed explanation of the flow-matching process in the revised version, particularly focusing on the sampling process.
>
> **5.Advantages of the flow-matching method itself**
> Thanks! While the advantages of the flow-matching method are indeed emphasized, they are directly relevant to social recommendation contexts. Specifically, flow-matching is particularly effective in handling the anisotropic noise structures commonly found in social graphs, which is central to our work. By better modeling these complex relationships, flow-matching improves the accuracy of recommendations in social networks.
>
> **6.How does flow matching specifically enhance performance within social recommendation scenarios?**
> Flow matching enhances performance in social recommendation by addressing two key challenges: structural noise and representation degradation caused by isotropic assumptions. Unlike diffusion-based models that rely on isotropic Gaussian noise, flow matching captures the anisotropic nature of social graphs, where user preferences vary across communities. It learns a deterministic velocity field to directly map noisy representations to clean ones, avoiding stochastic sampling and preserving fine-grained patterns.
>
> **7. Time complexities in the experimental section.**
> Thanks! We will include this comparison in the revised version regarding training time.
> | Model    | Ciao | Epinions |
> |----------|------|----------|
> | RecDiff  | 4.5s | 9.1s     |
> | RecFlow  | 4.3s | 8.7s     |
>
>
> **8.Recall rate metrics.**
> Thanks！ Recall rate metrics are already included in the primary experimental results of the paper. These metrics provide a comprehensive evaluation of the model's performance in terms of its ability to baselines social recommendations.
>
> **9. Figures, color scheme, coordinate system distorted**
> Thanks! In the revised version, we will make an effort to modify the figure for better clarity.
>
> **10. Different sampling methods**
> Thanks! We agree that the derivation of the ODE equations requires more detailed mathematical treatment. Moreover, within the experimental section or the ablation study, we plan to visually illustrate the essential steps of the ODE derivation and explicitly demonstrate their impacts on model performance (primarily focusing on the Euler and Runge-Kutta methods).
> | Model        | Ciao Recall | Ciao NDCG | Epinions Recall | Epinions NDCG |
> |--------------|-------------|-----------|-----------------|---------------|
> | Euler        | 0.725       | 0.438     | 0.486           | 0.341         |
> | Runge-Kutta  | 0.720       | 0.435     | 0.483           | 0.339         |
>
>
> **11.Whether applying generative models to both the user and bipartite graph**
> Thanks！it's important to note that the bipartite graph and the social graph have different characteristics. In particular, the bipartite graph, which typically models user-item interactions, does not necessarily exhibit the same anisotropic noise distribution as the social graph, which captures more complex relationships between users or between items with varying strengths. Thus, the assumption of anisotropic noise may not always hold in bipartite graph scenarios.
> We have not yet explored this direction of a unified generative framework specifically for bipartite graphs, but we recognize the potential value of doing so. Our current focus is on modeling anisotropic noise within social graphs, and we plan to investigate the applicability of generative models to bipartite graphs in future work.

---

### Official Review · Reviewer_jYLr · 2025-03-10

**Overall Recommendation:** 3

**Summary:**

The study presents Recflow, a flow-matching model for social recommendation systems, which addresses challenges in traditional recommendation methods, especially in noisy social networks. The key issue with many graph-based approaches is their inability to handle noisy edges in social graphs, which can degrade performance. The study highlights the effectiveness of using flow-matching models to capture anisotropic characteristics of social data, unlike traditional isotropic Gaussian noise used in diffusion models, which can obscure relational structures. Moreover, the study offers a detailed comparison of RecFlow with other generative models like DDPM mathmatically, demonstrating its superior ability to handle the anisotropy inherent in social data. The computational efficiency is also highlighted, with RecFlow requiring fewer steps for convergence compared to traditional diffusion models.

**Claims And Evidence:**

Although the paper provides detailed experiments, first analyzing the dataset to demonstrate the inherent anisotropy of social data, and later using visualization methods to show that flow matching can partially match this anisotropy, I believe that a more theoretical definition and expression of isotropy and anisotropy, as discussed in the paper, are needed. Unfortunately, the theoretical foundation and mathematical descriptions in the paper are not sufficiently developed.

**Essential References Not Discussed:**

No.

**Experimental Designs Or Analyses:**

As mentioned earlier, the derivation process of the ODE equation must include a mathematical procedure, which needs to be explained in detail in the both methd and experiment section. If there is no visualized process, this part should at least be presented in the ablation study.

**Methods And Evaluation Criteria:**

I think it is meaningful to evaluate the method proposed in the paper using graph-based recommendation datasets and commonly used metrics in recommendation systems.

**Other Comments Or Suggestions:**

I believe it would be beneficial to compare this approach with other denoising techniques, such as some heuristic denoising methods, to explain why flow matching performs better. Additionally, it would be even more helpful if a case study could be included to illustrate how different types of noise (isotropic and anisotropic) affect performance.

**Other Strengths And Weaknesses:**

The study introduces a flow-matching diffusion model for social recommendation systems, highlighting the potential of generative models in capturing user interactions and social graph structures. Through extensive experiments, it demonstrates RecFlow's significant advantages in improving recommendation accuracy and diversity compared to existing models.
In addition, I believe the color scheme of the figures in the paper is inconsistent, and it doesn't align well with the main content. Some explanatory notes should be added to enhance the readability of the article.

**Questions For Authors:**

Why was the method based on stochastic differential equations (SDEs) not adopted in this context?

**Relation To Broader Scientific Literature:**

The latest work by [1] suggests that diffusion models do not necessarily require the noise addition phase.


[1] Sun, Qiao, et al. “Is Noise Conditioning Necessary for Denoising Generative Models?” arXiv, 2025.

**Theoretical Claims:**

As mentioned in the claims, I believe that the core argument of the paper, the so-called "isotropic" and "anisotropic" noise, requires a more rigorous mathematical definition. Additionally, I think the derivation process of the ODE equation should be more detailed, and this should also be fully reflected in the experimental section.

---

> ### Author Rebuttal · Authors · 2025-04-01
>
> **1. More theoretical foundation and mathematical descriptions.**
> Thanks, and we agree. Following most works [1][2], we also define the isotropy as noise obeys the standard normal distribution N(0,1), which leads to the uniform properties in all directions. While, anisotropy refers to noise distributions that deviate from such uniformity. For example, Gaussian noise with a non-diagonal covariance matrix introduces variability across different directions in the data space. In the revised version, we will provide rigorous mathematical characterizations of these concepts.
>
> [1] A directional diffusion graph transformer for recommendation
> [2] Denoising Diffusion Probabilistic Models
>
> **2. More detailed derivation of the ODE equation, and correlated experiments should be added.**
> Thanks! We agree, and we present it as follows, which will also be added in our revised paper.
> Euler method:
> $$ y_{n+1} = y_n + h f(t_n, y_n) $$
>
> Runge-Kutta method:
> $$ k_1 = h f(t_n, y_n), \quad k_2 = h f(t_n + \frac{h}{2}, y_n + \frac{k_1}{2}) $$
> $$ k_3 = h f(t_n + \frac{h}{2}, y_n + \frac{k_2}{2}), \quad k_4 = h f(t_n + h, y_n + k_3) $$
> $$ y_{n+1} = y_n + \frac{1}{6}(k_1 + 2k_2 + 2k_3 + k_4) $$
>
>
> Moreover, we plan to visually illustrate the essential steps of the ODE derivation and explicitly demonstrate their impacts on model performance (primarily focusing on the Euler and Runge-Kutta methods).
> | Model        | Ciao Recall | Ciao NDCG | Epinions Recall | Epinions NDCG |
> |--------------|-------------|-----------|-----------------|---------------|
> | Euler        | 0.725       | 0.438     | 0.486           | 0.341         |
> | Runge-Kutta  | 0.720       | 0.435     | 0.483           | 0.339         |
>
>
> **3.[1] suggests that diffusion models do not necessarily require the noise addition phase**.
> Thanks! What [1] suggests is indeed highly relevant to our RecFlow. In essence, RecFlow aligns closely with the perspective: rather than relying solely on the standardized noise addition typical of traditional diffusion models. Concretely, RecFlow employs a flexible flow-matching mechanism, avoiding the masking of relational structures caused by standard isotropic noise. This allows RecFlow to preserve relational information more accurately. Thus, our research provides complementary evidence supporting the assertion of [1] that diffusion models need not strictly rely on traditional isotropic noise. We will explicitly discuss this alignment in the next revision.
>
> [1]Is Noise Conditioning Necessary for Denoising Generative Models?
>
> **4. Color Scheme**
> Thanks! We will consider adjusting the figure in the revised version to make it clearer.
>
> **5. Comparison with other heuristic denoising methods.**
> Thanks! We have already made a comparison with some traditional denoising models like DSL [1], and the results show that the generative model should ideally provide more accurate and diverse predictions because it can better capture the patterns in the noisy data. Additionally, we agree that comparing with heuristic denoising methods would be beneficial, and we plan to include such a comparison in future work. As for the impact of different types of noise, we will incorporate a case study to illustrate how isotropic and anisotropic noise affect model performance, which will further highlight the advantages of our flow-matching approach[2]
> | Model    | Ciao Recall | Ciao NDCG | Epinions Recall | Epinions NDCG |
> |----------|-------------|-----------|-----------------|---------------|
> | GDMSR    | 0.560       | 0.355     | 0.368           | 0.241         |
> | DSL      | 0.606       | 0.389     | 0.365           | 0.267         |
> | RecFlow  | 0.725       | 0.438     | 0.486           | 0.341         |
>
>
> [1]Denoised Self-Augmented Learning for Social Recommendation
> [2]Denoising Diffusion Probabilistic Models
>
> **6. More discussion on stochastic differential equations (SDEs) based methods.**
> Thanks! There are existing works [1] using SDE approaches for social recommendation. Additionally, SDE methods predominantly rely on isotropic noise, making them inadequate for effectively capturing the anisotropic characteristics inherent in the social networks emphasized in our study. We will add these discussion in the final version.
> [1] Score-based Generative Diffusion Models for Social Recommendations

---

> > ### Comment · Reviewer_jYLr · 2025-04-05
> >
> > Thank you for answering my questions. I think the author has addressed my concerns.

---

### Official Review · Reviewer_Ec7S · 2025-03-14

**Overall Recommendation:** 3

**Summary:**

This paper introduces RecFlow, a social recommendation system using flow matching to handle noise in social graphs. Unlike traditional diffusion models that assume isotropic noise, RecFlow better captures the anisotropic nature of social data by learning velocity fields that preserve structural relationships. Experiments across three datasets show RecFlow outperforms state-of-the-art methods, with 2-5.6% improvements in Recall and 4.5-10.7% in NDCG metrics. The approach requires fewer computational steps than traditional diffusion models while maintaining better data structure preservation.

**Claims And Evidence:**

a) As far as I know, the errors in social networks may include label errors (i.e., annotation mistakes) and some outdated or irrelevant connections.
b) The paper mentions denoising, but it's unclear which type of noise this refers to, or if it addresses other kinds of noise.
c) There is no detailed explanation provided in the paper. Additionally, why is the denoising process limited to the social network, while noise can also exist on the user-item edges? Why isn't denoising applied to the user-item bipartite graph as well?

**Essential References Not Discussed:**

Liu, C., Zhang, J., Wang, S., Fan, W., & Li, Q. (2024). Score-based Generative Diffusion Models for Social Recommendations.

**Experimental Designs Or Analyses:**

The paper includes extensive experiments covering performance, ablation, visualization, convergence, and sensitivity analysis

However, one question remains: how is the method demonstrated to be robust against newly introduced noise? Furthermore, how exactly is the noise removed? This aspect should also be addressed in the experiments.

**Methods And Evaluation Criteria:**

There is a score-based work from 2024: Liu, C., Zhang, J., Wang, S., Fan, W., & Li, Q. (2024). Score-based Generative Diffusion Models for Social Recommendations. It also applies generative models to social recommendations, but no comparison is made in the paper.

**Other Comments Or Suggestions:**

N/A

**Other Strengths And Weaknesses:**

Strengths:
1. The proposed RecFlow method introduces a novel flow matching approach to social recommendation that effectively handles anisotropic noise in social graphs. This represents a significant advancement over traditional diffusion models that assume isotropic Gaussian noise.
2. The experimental evaluation is comprehensive, covering three datasets and comparing against 12 baselines. The results show consistent improvements (2-5.6% in Recall and 4.5-10.7% in NDCG) over state-of-the-art methods, with thorough ablation studies and analyses of robustness, sensitivity, and convergence.

Weaknesses:
1. The paper lacks discussion of existing score-based social recommendation methods, creating a gap in the literature review. This omission makes it difficult to fully contextualize the contribution within the broader landscape of generative approaches to social recommendation.
2. Despite focusing on denoising, the paper provides insufficient explanation of the specific types of noise being addressed in social graphs. While it mentions "graph-level redundancy" and "graph-level missing," it doesn't clearly define these concepts or explain how they manifest in real-world data, limiting understanding of the method's practical applications.

**Questions For Authors:**

Please refer to my comments above.

**Relation To Broader Scientific Literature:**

Previous work has already explored score-based methods for social diffusion, and I’ve provided the paper title in the Evaluation Criteria section.

Given methods like DDPM and score-based models, which are based on SDEs, what advantages does flow matching, which is based on ODEs, offer in comparison?

**Theoretical Claims:**

How is the supervisory signal introduced in the generative model in the paper?
The paper doesn't provide a detailed explanation, or at least I didn't fully understand it.

---

> ### Author Rebuttal · Authors · 2025-04-01
>
> **1. It is unclear whether it addresses label errors, outdated connections, or other types of noise.**
> The noise we primarily address is graph-based noise, where social edges may represent outdated or misleading connections, potentially degrading the quality of recommendations. Regarding the type of noise mentioned, the paper focuses on anisotropic noise in social graphs. This refers to the varying structure of relationships across different parts of the network, where some edges might be more meaningful or relevant than others. The anisotropic nature of this noise distinguishes it from the isotropic Gaussian noise commonly assumed in other models.
>
>
> **2. Why denoising is limited to the social network, ignoring noise on user-item bipartite graph.**
> The characteristics of bipartite graphs and social graphs differ significantly; while bipartite graphs typically model user-item interactions, social graphs capture more complex relationships between users or items with varying strengths. It is important to note that bipartite graphs may not necessarily exhibit the same anisotropic noise distribution as social graphs. Although we have not yet explored a unified generative framework specifically for bipartite graphs, we recognize its potential value. Our current focus is on modeling anisotropic noise within social graphs, with plans to investigate the applicability of generative models to bipartite graphs in future research.
>
> **3. More discussion on stochastic differential equations (SDEs) based methods.**
> Thanks! There are existing works [1] using SDE approaches for social recommendation. Additionally, SDE methods predominantly rely on isotropic noise, making them inadequate for effectively capturing the anisotropic characteristics inherent in the social networks emphasized in our study. We will add these discussion in the final version.
> [1] Score-based Generative Diffusion Models for Social Recommendations
>
>
> **4. Supervisory signals?**
> The supervisory signal mentioned refers to the label, which acts as a condition in the model. These labels guide the interpretation and application of the noise model, specifying the characteristics of the noise to be modeled in each scenario. By conditioning the system on these labels, we ensure that the model adapts its behavior according to the specific type of noise or variation present in the data. The labels allow the model to differentiate between different data distributions or contexts, improving its ability to make accurate predictions. In the revised version, we will further elaborate on the role of these labels in conditioning the system.
>
> **5. Robustness?**
> We have already conducted a robustness analysis, where we examine the impact of the noise scale factor (τ) on the noising process. As the noise scale decreases from 1 to 0.1, model performance improves, with higher Recall@20 and NDCG@20 values for both Yelp and Epinions, demonstrating the effectiveness of RecDiff's denoising mechanism. However, when the noise scale reaches a certain threshold (τ = 10−2 and 10−3), excessive noise causes performance degradation, particularly in NDCG@20, as too much noise interferes with the model's ability to retain important user-item information.
>
> **6. Lacks discussion of existing score-based social recommendation methods**
> Existing works like [1] apply SDEs to recommendation, akin to Q2, but use isotropic noise, limiting their capture of social networks' anisotropic traits, highlighted in our study. Anisotropic noise, reflecting community patterns interactions, is key to understanding social graphs. We'll detail these SDE methods in the final version's Related Work section for better context.
> As for the comparison, we primarily focus on Ciao and Epinions, as SGSR does not include a comparison with Yelp.
> | Model     | Recall (Epinions) | NDCG (Epinions) | Recall (SGSR) | NDCG (SGSR) |
> |-----------|-------------------|-----------------|---------------|-------------|
> | SGSR      | 0.645             | 0.425           | 0.470         | 0.315       |
> | RecFlow   | 0.725             | 0.438           | 0.486         | 0.341       |
>
> [1]Score-based Generative Diffusion Models for Social Recommendations
>
> **7. Lacks a clear explanation of the specific types of noise in social graphs**
>
> Thanks! We agree that the explanation of the specific types of noise in social graphs could be more detailed. The noise we primarily address is graph-based noise, where social edges may represent outdated or misleading connections, potentially degrading the quality of recommendations. Regarding the type of noise mentioned, the paper focuses on anisotropic noise in social graphs. This refers to the varying structure of relationships across different parts of the network, where some edges might be more meaningful or relevant than others. The anisotropic nature of this noise distinguishes it from the isotropic Gaussian noise commonly assumed in other models.

---

> > ### Comment · Reviewer_Ec7S · 2025-04-02
> >
> > Thank you for your detailed responses to my review, which addressed my concerns. Therefore, I will raise my score.

---

### Decision · Program_Chairs · 2025-05-01

**Decision:**

Accept (poster)

**Comment:**

All reviewers give positive scores. The paper is well-structured and the experimental evaluation is comprehensive. Extensive performances prove the promising performances of our RecFlow from six aspects, including superiority, effectiveness, robustnesses, sensitivity, convergence and visualization.